# Assessing independence in mobility activities in trauma care: Validity and reliability of the Activity Independence Measure-Trauma (AIM-T) in humanitarian settings

Bérangère Gohy[1,2]*, Christina H. Opava[1], Johan von Schreeb[3], Rafael Van den Bergh[4], Aude Brus[5], Nicole Fouda Mbarga[6], Jean Patrick Ouamba[6], Jean-Marie Mafuko[7], Irene Mulombwe Musambi[8], Delphine Rougeon[8], Evelyne Côté Grenier[9], Lívia Gaspar Fernandes[9], Julie Van Hulse[10], Eric Weerts[2], The AIM-T Study Group[4,6,7,8,9,10¶], Nina Brodin[1,11]

1 Department of Neurobiology, Division of Physiotherapy, Care Sciences and Society, Karolinska Institutet, Stockholm, Sweden, 2 Humanity & Inclusion, Rehabilitation Technical Direction, Brussels, Belgium, 3 Department of Global Public Health, Karolinska Institutet, Stockholm Sweden, 4 Médecins Sans Frontières, Operational Center Brussels, Brussels, Belgium, 5 Humanity & Inclusion, Innovation, Impact & Information Division, Brussels, Belgium, 6 Médecins Sans Frontières, Operational Center Geneva, YaoundeYaounde, Cameroon, 7 Médecins Sans Frontières, Operational Center Brussels, Bujumbura, Burundi, 8 Médecins Sans Frontières, Operational Center Paris, Bangui, Central African Republic, Baghdad, Iraq, 9 Médecins Sans Frontières, Operational Center Paris, Baghdad, Iraq, 10 Médecins Sans Frontières, Operational Center Paris, France, 11 Department of Orthopaedics, Danderyd Hospital Corp., Division of Physiotherapy, Danderyd, Sweden

¶ Membership of the AIM-T study group is provided in the Acknowledgments
* berangere.gohy@ki.se

## Abstract

The importance of measuring outcomes after injury beyond mortality and morbidity is increasingly recognized, though underreported in humanitarian settings. To address short-comings of existing outcome measures in humanitarian settings, the Activity Independence Measure-Trauma (AIM-T) was developed, and is structured in three subscales (i.e., core, lower limb, and upper limb). This study aimed to assess the AIM-T construct validity (structural validity and hypothesis testing) and reliability (internal consistency, inter-rater reliability and measurement error) in four humanitarian settings (Burundi, Iraq, Cameroon and Central African Republic). Patients with acute injury (n = 195) were assessed using the AIM-T, the Barthel Index (BI), and two pain scores. Structural validity was assessed through confirmatory factor analysis. Hypotheses were tested regarding correlations with BI and pain scores using Pearson correlation coefficient (PCC) and differences in AIM-T scores between patients' subgroups, using standardized effect size Cohen's d (d). Internal consistency was assessed with Cronbach's alpha (α). AIM-T was reassessed by a second rater in 77 participants to test inter-rater reliability using intraclass correlation coefficient (ICC). The results showed that the AIM-T structure in three subscales had an acceptable fit. The AIM-T showed an inverse weak to moderate correlation with both pain scores (PCC<0.7, p≤0.05), positive strong correlation with BI (PCC≥0.7, p≤0.05), and differed between all subgroups (d≥0.5, p≤0.05). The inter-rater reliability in the (sub)scales was good to excellent (ICC

**Data Availability Statement:** The authors confirm that, for approved reasons, some access restrictions apply to the data underlying the findings. Due to the sensitive nature of trauma care (including violent trauma) data, full datasets are not made available by default. Data are available through the MSF Data Sharing Agreement for researchers who meet the criteria for access to confidential data; requests should be addressed to the Data Sharing Agreement coordinator, Annick Antierens (Annick.Antierens@brussels.msf.org).

**Funding:** Elrha's Research for Health in Humanitarian Crises (R2HC) programme (Grant No.32398) funded the research coordination (BG) and research activity for this study, funded by the UK Foreign, Commonwealth and Development Office (FCDO), Wellcome Trust, and the UK National Institute for Health Research (NIHR) (https://www.elrha.org/programme/research-for-health-in-humanitarian-crises/). The funders had no role in study design, data collection and analysis, decision to publish, or preparation of the manuscript.

**Competing interests:** The authors have declared that no competing interests exist.

0.86–0.91) and the three subscales' internal consistency was adequate (α≥0.7). In conclusion, this study supports the AIM-T validity in measuring independence in mobility activities and its reliability in humanitarian settings, as well as it informs on its interpretability. Thus, the AIM-T could be a valuable measure to assess outcomes after injury in humanitarian settings.

## Introduction

Injury accounts for 4.3 million deaths yearly and represented 9.8% of the global burden of disease in 2019 [1]. While the years of life lost (YLLs) rate due to injury has decreased by 23.6% between 2010 and 2019, years lived with disability (YLDs) have increased by 1.1% over the same time period [2]. In low- and middle-income countries (LMICs), as well as in humanitarian settings, the burden of injury is high, though its extent is underestimated and underreported [3–6]. Humanitarian settings are defined here as "situations in which there is a widespread threat to life, physical safety, health or basic subsistence that is beyond the coping capacity of individuals and communities in which they reside", due to chronic or sudden-onset crises, caused by natural or technological disasters, famine, epidemics or armed conflict [7]. In such settings, persons living with disability, either resulting from injury or from other health conditions, may face additional barriers in accessing basic needs (e.g., food or shelter), eventually suffering greater social and economic consequences [8–10]. The implementation of organized trauma care has significantly decreased mortality rates after injury [11]. Trauma care should however also strive to prevent avoidable disability in humanitarian settings [3].

Similarly, indicators used for trauma care monitoring have mainly focused on hospital processes and morbidity and mortality outcomes rather than on disability [5,12–15]. However, functioning has increasingly been recommended as a quality indicator of trauma care [14,16–19]. Full functioning and complete disability are extremes of the same continuum, capturing the "dynamic interaction between a person's health condition, environmental factors and personal factors" [20,21]. Although functioning and disability need to be comprehensively understood, targeted assessments of specific domains are recommended for patient-centered care [22,23]. Mobility and self-care activities are frequently limited after injury, and regaining independence in these activities is among the patients' priorities [24]. Their measurement is therefore an essential part of daily clinical practice, while also being useful for trauma care monitoring. In particular, the use of measures of independence in activities based on observation may increase opportunities to perform such activities within routine care, thereby encouraging early mobilization, which is a common challenge in humanitarian settings [19,25,26].

Several measures have been used to assess independence in activities after injury, including Barthel Index (BI), Functional Independence Measure (FIM), Short Musculoskeletal Function Assessment Questionnaire (SMFA), Short Form health survey–Physical function (SF36-PF), Activity Measure for Post-Acute Care (AM-PAC) and Patient-Reported Outcome Measurement Information System–Physical function (PROMIS-PF)) [27–32]. Most of these have been developed and assessed in non-humanitarian settings. Some measures are self-reported, which may be impractical in settings with low literacy rate [33]. Also, lengthy questionnaires, accredited administration or license fees, lack of available versions in local languages, or lack of culturally adapted content are barriers that limit application of existing measures across humanitarian settings and hinder comparability of results [33–37]. Furthermore, in heterogenous populations, such as patients after injury, the use of multiple specific measures hinders comparison between subgroups and increases clinician workload [38]. The Activity

Independence Measure-Trauma (AIM-T) has the potential of addressing a number of shortcomings of existing measures, being a generic and observed measure designed for and used in humanitarian settings. A preliminary version of this measure was described in 2016 as part of retrospective study in Afghanistan and was subsequently revised to address time constraints, cultural relevance, and appropriateness, based on content validity assessment [39,40]. The quality of the information produced by a measure depends on its measurement properties (i.e., validity and reliability) in the context where it is used [41]. To further support the AIM-T validity as a measure of independence and its reliable use across health care professionals and health structures, our study aimed to assess the AIM-T construct validity (structural validity and hypothesis testing) and reliability (internal consistency, inter-rater reliability, and measurement error) among patients admitted for trauma care in four humanitarian settings. Testing the AIM-T measurement properties in its context of use allows informed choices by potential users, eventually fostering the reporting of functioning after injury in such settings.

## Materials and methods

This cross-sectional study was conducted between July 2019 and November 2019 and tested the revised version of the AIM-T, according to the Consensus-based Standards for the selection of health Measurement Instruments framework (COSMIN) [42].

### Study setting

We collected data from centres supported or run by the medical non-governmental organisation Médecins Sans Frontières (MSF), which provides trauma care in humanitarian settings. The selection of the four centres was based on their geographic diversity and their capacity to collect data.

Data from the following centres were included in this study:

- The MSF trauma centres of Arche (Bujumbura, Burundi) and Sica (Bangui, Central African Republic), which are set up following MSF trauma centre standards [43,44].

- The Regional Hospital of Maroua (Cameroon), where MSF supported the surgical department for urgent surgical cases.

- The MSF Baghdad Medical Rehabilitation Centre (Iraq), providing post-operative care.

In- and outpatient physiotherapy was provided in all four centres.

### Study population

We aimed to include 50 consecutive patients in each of the four centres. Patients aged five years or above, within six months of their injury, and receiving in- or outpatient physiotherapy service were eligible. To ensure sufficiently large samples for each sub-group analysis, we balanced between centres receiving more patients with acute (i.e., within 30 days after injury) versus post-acute injury (i.e., between 31 days and 6 months after injury). Orthopaedic, visceral, and soft tissue injuries were grouped by location: lower limb and pelvis, upper limb, and trunk (i.e., spine, abdomen, and chest). Patients with isolated central neurologic injuries were excluded.

For inter-rater reliability assessment, a subsample of 20 patients was purposively selected in each centre for a second assessment. Selection aimed at diversity in terms of sex and age, as well as location, nature, severity, and acuteness of injury. Two raters were recruited from each centre. All raters were trained physiotherapists working in the study centres.

All participants consented to participate in the study. For patients under 18 years old or having cognitive difficulties, a legal representant gave written informed consent. Patients between 12 and 18 were also requested to provide their written informed assent to participate in the study. A witness was used to confirm verbal consent for patients with low literacy levels as well as for patients volunteering for the study but unable to give written consent.

## Measurements

Routine and study-specific data were collected in the four study centres. Routine data included demographic information (age and sex), and clinical data on the injuries (date, location, nature, severity). The South African Triage Score (SATS) categorises the injury severity by colour, from minor injuries, labelled as 'green', to emergency to be seen immediately, labelled as 'red' [45–47]. The SATS was routinely scored and used as a proxy for injury severity in this study. Study-specific data comprised:

The *Activity Independence Measure–Trauma (AIM-T)* is composed of 12 activities grouped into three subscales (i.e., core, lower limb, upper limb), ranging from 0 to 10 for the core subscale and 0 to 25 for the lower limb and the upper limb subscales. The total score ranges from 0 to 60, based on the difficulties observed and level of human or material assistance required (higher score indicating a higher independence in activities) (S1 Fig).

The *Barthel Index (BI)* is a generic measurement of independence in activities, used for self-report or clinician rating [27]. It is composed of ten activities, with a total score ranging from 0 to 100, depending on the level of assistance required (higher score indicating a higher level of independence). The BI has been used in patients after injury, including in humanitarian settings [48–52]. The French and Arabic validated versions were used and were revised following a linguistic validation process instructed by BI license holders.

The *Visual Analogue Scale (VAS) and Faces Pain Scale-revised (FPS-R)* are self-reported pain measurements previously used with patients after injury [53,54]. To accommodate cultural and literacy aspects, patients were given the choice between the two scales [55–58]. Using either of these scales, pain was evaluated both at rest and during one activity chosen by each patient. The FPS-R scores were converted to equal VAS scores with "10" representing the worst pain and "0" no pain [57].

## Procedure

An eight-hour rater training was provided remotely by the first author in French for the centres in Burundi, Cameroon and CAR, and in English translated to Arabic for the centre in Iraq. It included instructions on the use of the different measurements, and two two-hour training sessions on the AIM-T, using videos of each activity throughout the trainings to reduce potential information loss due to translation. A written guideline describing in detail the AIM-T activities and scoring system was also provided, together with feedback sessions on its use with pilot patients. Sessions were organised joining several centres when possible.

Study-specific data were collected by one rater in each centre, assessing, in sequential order, the included patients with the AIM-T, the BI, and either of the pain scales. The administration of the AIM-T and BI was timed and both were administered as observed measures. Self-reporting for BI was used only for patients refusing to perform any of its activities [59]. The mode of administration was documented for each of the BI activities.

To assess inter-rater reliability, the selected patients were re-assessed with the AIM-T by a second rater blinded to the first assessment results. A 30-minute rest was allowed between the two assessments. Test conditions were similar for both assessments in terms of equipment, instructions, and planned physiotherapy session.

The raters also documented any other relevant observation regarding the assessment conditions or patient health status (e.g., refusal to perform any activity or pain reported during the activity).

## Data analyses

Data are described using frequencies, medians with interquartile ranges (IQR), or means and standard deviations (SD). All quantitative analyses were performed in SPSS version 27.

The construct validity of the AIM-T was assessed through *structural validity* and *hypothesis testing*. All hypotheses were formulated before data analysis. Confirmation of at least 75% of the hypotheses is considered sufficient to support construct validity [60].

S*tructural validity* was investigated through confirmatory factor analysis (CFA). As suggested by the AIM-T content validity assessment, clinical reasoning, and literature, we hypothesized that a three-factor model, which represent the three AIM-T subscales (i.e., core, lower limb and upper limb activities), would better fit the items of the AIM-T than a single-factor model [39,61]. The three-factor and the single-factor models were both tested, using CFA in Jamovi 2.2.5. Model fits were assessed using three fit indices and their goodness of fit criteria: root mean square error of approximation (RMSEA) ($<0.06$), comparative fit index (CFI) ($>0.95$), and standardized root mean square residuals (SRMR) ($\leq 0.08$) [62]. At least one of these criteria should be met to support the structural validity [62]. The unidimensionality of the subscales was assessed using Principal Component Analysis (PCA). The (sub)scale was considered unidimensional when 1) the first factor explained $\geq 20\%$ of the variance and 2) the ratio of the variance explained by the first factor to the variance explained by the second factor is greater than four [63].

For *hypothesis testing*, we investigated *known-group validity* and *concurrent validity*. For *known-group validity*, hypotheses were formulated based on clinical experience and known associations of covariates (i.e., acuteness, location and number of injuries) with independence in activities after injury [64–68]. Patients' subgroups were expected to differ in AIM-T scores by a Cohen's d standardised effect sizes larger or equal to 0.5 [69]. For *concurrent validity*, hypotheses were tested using Pearson correlation coefficient (PCC) [70]. We hypothesized a weak to moderate negative correlation between the AIM-T and pain scores (PCC < -0.7), stronger when assessed during activity. Although pain and independence in activities are distinct constructs, they are commonly associated after injury, with pain interfering with daily activities [71,72]. We hypothesised a strong positive correlation (PCC $\geq$ 0.7) between BI and the AIM-T because they intend to measure the same construct. The reliability of the AIM-T was assessed through *internal consistency*, *inter-rater reliability* and *measurement error*.

For *internal consistency*, a Cronbach alpha coefficient ($\alpha$) was calculated for each (sub)scale having a confirmed unidimensionality in CFA. A coefficient equal to or larger than 0.70 was considered adequate [62]. If lower than 0.70, the effect of deleting separate items from the AIM-T to reach a higher $\alpha$ was investigated [73].

The *inter-rater reliability* was determined using the intraclass correlation coefficients (ICC) between pooled scores from the first and second raters. The ICC for absolute agreement was calculated for each (sub)scale, based on a two-way mixed effect [73]. Poor, moderate, good and excellent reliability were indicated by an ICC of less than 0.5, between 0.5 and 0.75, between 0.75 and 0.9, and greater than 0.9, respectively [74]. An ICC greater than 0.7 indicating sufficient reliability based on COSMIN guidance [62,74]. Patients who refused to perform at least one activity were removed from the inter-rater reliability analysis.

*Measurement error* was calculated as the standard error of measurement (SEM), the smallest detectable change (SDC) and the limits of agreement (LoA) for the total score and subscales

scores. SEM$_{agreement}$ was derived from the ICC, as the root square of the error variance [73]. The SDC was calculated as 1.96 x SD$_{difference}$. For the LoA, a Bland and Altman plot was performed, which plots the mean difference between the first and the second raters against the pooled AIM-T means from the two raters [73,75]. Upper and lower LoA are defined as: Mean-$_{difference}$ ± SDC. The proportion of the mean differences being within the limits of agreement was assessed, as well as the heteroscedasticity of the variables [73,76].

## Ethics

The protocol of this study was approved by the MSF Ethics Review Board, Geneva, Switzerland (reference ID 1893) and the Swedish Ethical Review Authority, Stockholm, Sweden (Dnr 2022-02806-01). National ethics review committees competent for each participating centre also approved the study, i.e., Burundi National Ethics Committee for the Protection of Human Rights of Participants in Biomedical and Behavioural Research (15/04/2019), Scientific Committee for the Validation of Study Protocols and Research Results on Health in Central African Republic (28/UB/FACSS/CSCVPER/19), the National Ethics Committee for Research in Human Health in Cameroon (2019/08/1184/CE/CNERSH/SP) and ethics committees of the Baghdad Directorate of Health in Iraq (p. 1/5/10).

## Results

### Participants

A total of 195 patients were included from the four study centres: 50 in Baghdad, 44 in Bangui, 50 in Bujumbura, and 51 in Maroua. Most patients were young males (66% male, 62% aged between 18 and 45 years). Most patients presented with single injury (75%) and at least one lower limb injury (75%), while less patients presented with upper limb injury (30%), trunk injury (10%) or both upper and lower limb injury (10%). The median time since injury was 36 days (IQR 5–83). The characteristics of the included patients are presented in Table 1.

The mean AIM-T total score was 44.2 (SD +/- 10.5), while the mean BI score was 80.9 (SD +/- 18.6). It took on average 10.9 (SD +/- 10.2) and 10.5 minutes (SD +/- 9.8) to administer the AIM-T and the BI, respectively. The assessment of the BI was mostly self-reported (66%), with self-care activities (i.e., 'getting on and off toilet', 'bathing self', and 'dressing and undressing') being self-reported by at least 80% of the patients, rather than observed. Patients reported significantly more intense pain during activity (VAS mean 4.2, SD +/- 2.1 and FPS-R mean 5.7, SD +/- 2.5) than at rest (VAS mean 1.9, SD +/- 1.8 and FPS-R mean 2.7, SD +/- 2.7), p<0.001.

### Validity

For *structural validity*, the three-factor model met the COSMIN fit criteria and fit better to this study sample than the single-factor model (Table 2). The visual representation of the three-subscale model and the coefficient of each item is available in S2 Fig. The PCA supported the unidimensionality of the three subscales but not of the total scale. The first factor of the (sub) scales explained at least 20% of the variance, ranging from 38.9% for the total scale to 85.7% for the core subscale. The ratio of variance was above four for the three subscales (core 5.9, lower limb 7.5, and upper limb 7.6) but not for the total scale (1.1).

For *hypothesis testing*, in terms of *known group validity*, all the patients' subgroups differed significantly in AIM-T total or subscale scores with a magnitude equal or superior to the hypothesised differences (d≥0.5). The *concurrent validity* of the AIM-T was supported by its correlation with all the comparator measurements with the strength hypothesized (PCC<0.7 for pain scales, and PCC≥0.7 for BI). AIM-T mean scores and standardised effect sizes per

**Table 1. Characteristics of the 195 patients assessed to test validity and reliability of the Activity Independence Measure–Trauma (AIM-T), and of the 77 patients among them who were assessed a second time to test inter-rater reliability.**

| Characteristics | Total sample n (%) | Inter-rater reliability sample n (%) |
|---|---|---|
| **Total number** | **195** | **77** |
| **Age, years** | | |
| 5–17 | 32 (16.4) | 13 (16.9) |
| 18–45 | 121 (62.1) | 42 (54.5) |
| >45 | 42 (21.5) | 22 (28.6) |
| **Sex** | | |
| Male | 129 (66.2) | 42 (54.5) |
| Female | 66 (33.8) | 35 (45.5) |
| **SATS[a] triage colour** | | |
| Green | 13 (6.7) | 10 (13.0) |
| Yellow | 88 (45.1) | 32 (41.5) |
| Orange | 43 (22.1) | 15 (19.5) |
| Red | 1 (0.5) | 0 (0.0) |
| Missing | 50 (25.6) | 20 (26.0) |
| **Injury acuteness** | | |
| < 30 days | 91 (46.7) | 36 (46.8) |
| 30 days—6 months | 104 (53.3) | 41 (53.2) |
| **Injury type[b]** | | |
| ≥ 1 lower limb fracture | 105 (53.5) | 41 (53.2) |
| ≥ 1 lower limb soft tissue injury | 70 (35.9) | 23 (29.9) |
| ≥ 1 lower limb amputation | 8 (4.1) | 5 (6.5) |
| ≥ 1 upper limb fracture | 41 (21.0) | 15 (19.5) |
| ≥ 1 upper limb soft tissue injury | 22 (11.8) | 6 (7.8) |
| ≥ 1 upper limb amputation | 4 (2.1) | 2 (2.6) |
| ≥ 1 visceral injury | 9 (4.6) | 3 (3.9) |
| ≥ 1 other injury | 28 (14.3) | 12 (15.6) |
| **Number of Injuries** | | |
| Single | 147 (75.4) | 60 (77.9) |
| Multiple | 48 (24.6) | 17 (22.1) |

[a]South African Triage Score (SATS),

[b]Patients could have more than one injury type.

**Table 2. Goodness of fit indices for the AIM-T three-factor and single-factor models, based on structural equation modelling, comparing the COSMIN fit criteria to the observed fit indices to test structural validity.**

| | RMSEA[a] | CFI[b] | SRMR[c] | COSMIN criteria fulfilled (≥ one goodness of fit) |
|---|---|---|---|---|
| Goodness of fit criteria [62] | <0.06 | >0.95 | <0.08 | NA |
| Three subscale Model | 0.14 | 0.90 | 0.07 | Yes |
| One factor Model | 0.31 | 0.47 | 0.26 | No |

[a]RMSEA = Root mean square error of approximation,

[b]CFI = Comparative fit index (CFI),

[c]Standardized root mean square residuals (SRMR).

**Table 3. AIM-T mean scores (SD) of the total sample and the different subgroups, comparing the hypothesized and observed effect sizes to test known group validity.**

| Subgroups | n | Hypothesized Cohen's d | AIM-T score, mean (SD) | Observed Cohen's d | p value | Hypothesis confirmed |
|---|---|---|---|---|---|---|
| **Total sample** | 195 | NA | **Total score:** 44.2 (10.5) | NA | NA | NA |
| | 195 | | **Core score:** 8.3 (2.3) | | | |
| | 195 | | **Lower limb score:** 14.5 (7.7) | | | |
| | 195 | | **Upper Limb score:** 21.4 (5.8) | | | |
| | | ≥0.5 | **Total score:** | 0.5 | 0.021 | Yes |
| **Acute injury** | 91 | | 41.4 (11.7) | | | |
| **Post-acute injury** | 104 | | 46.6 (8.6) | | | |
| | | ≥0.5 | **Lower limb score:** | 1.3 | 0.047 | Yes |
| **≥ 1 lower limb injury** | 147 | | 11.9 (6.5) | | | |
| **No lower limb injury** | 48 | | 22.2 (5.8) | | | |
| | | ≥0.5 | **Upper limb score:** | 1.3 | <0.001 | Yes |
| **≥ 1 upper limb injury** | 59 | | 16.0 (6.9) | | | |
| **No upper limb injury** | 136 | | 23.8 (3.2) | | | |
| | | ≥0.5 | **Total score:** | 0.7 | <0.001 | Yes |
| **Single** | 147 | | 46.0 (8.9) | | | |
| **Multiple** | 48 | | 38.7 (12.8) | | | |

patients' subgroups are presented in Table 3, and the AIM-T correlation with comparator measurements in Table 4.

## Reliability

The *internal consistency* was adequate for the three unidimensional subscales (α 0.82 for core, 0.92 for lower limb and 0.91 upper limb subscales respectively).

For *inter-rater reliability* testing, 6 (all from Bangui) of 83 re-assessed patients were removed from the analysis because they had refused to perform at least one activity during the second assessment. The remaining 77 patients were mostly young males with single limb injuries (Table 1). The mean AIM-T scores for the second raters was 41.9 (SD +/- 11.2), differing

**Table 4. Correlation between the AIM-T and the Barthel Index and pain scores, compared with the hypothesized correlations to test concurrent validity.**

| Comparator measurement | n | Hypothesized correlation, r | Observed correlation, r | p value | 95% Confidence interval | Hypothesis confirmed |
|---|---|---|---|---|---|---|
| **Barthel Index** | 195 | ≥ 0.7 | 0.83 | <0.001 | 0.78–0.87 | Yes |
| **Pain at rest** | | | | | | |
| VAS[a] | 96 | < -0.7 | -0.26 | 0.010 | -0.44 – -0.06 | Yes |
| FPS-R[b] | 100 | < -0.7 | -0.47 | <0.001 | -0.60 – -0.29 | Yes |
| **Pain during activity** | | | | | | |
| VAS[a] | 89 | < -0.7 and stronger than pain at rest | -0.32 | 0.002 | -0.50 – -0.12 | Yes |
| FPS-R[b] | 100 | | -0.52 | <0.001 | -0.65 – -0.36 | Yes |

[a]VAS = Visual analogue scale,

[b]FPS-R = Faces pain scale–revised.

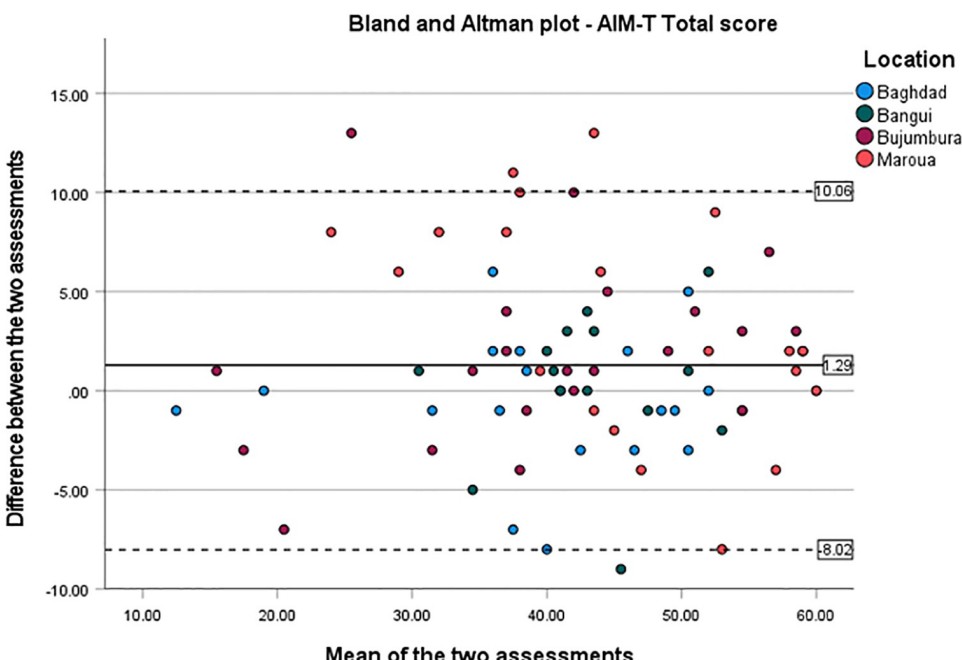

**Fig 1. Bland and Altman plot for the AIM-T total score.** Differences in AIM-T total scores between the first and the second raters are plotted against the pooled mean of the AIM-T scores for patients after injury in four humanitarian settings (n = 77). Mean difference between the two raters (-1.29) is represented by a solid line, while the limits of agreement (-8.02 to 10.07) by dashed lines.

significantly from the first raters on average by -1.3 (SD +/- 4.6) (p = 0.008, CI 0.2 to 2.3). The inter-rater reliability was excellent for the total scale (ICC 0.91, CI 0.85 to 0.94) and the lower limb subscale (ICC 0.91, CI 0.86 to 0.94), and good for the core (ICC 0.86, CI 0.79 to 0.91), and upper limb (ICC 0.88, CI 0.81 to 0.92) subscales.

Regarding *measurement error*, the $SEM_{agreement}$ was 3.36, 0.96, 2.41, and 1.88 for the AIM-T total, core, lower limb, and upper limb (sub)scales respectively. The SDC was 9.31, 2.66, 6.68, and 5.19 for the total, core, upper limb, and lower limb (sub)scales. The LoA ranged from -8.02 to 10.07 for the AIM-T total scale. As shown on the Bland and Altman plot, the measurement error between raters does not vary depending on the AIM-T scores, indicating no heteroscedasticity (Fig 1). The Bland and Altman plots for the subscales can be found in S3 Fig.

## Discussion

This study supports the validity and reliability of the AIM-T in measuring independence in activities of patients after injury in four different humanitarian settings. First, different aspects of validity of the AIM-T are supported based on the consistence with pre-specified hypotheses concerning its structure, subgroup differences, and correlation with other validated measurements, exceeding the 75% COSMIN criteria [60]. Second, the AIM-T demonstrated good to excellent inter-rater reliability and adequate internal consistency, meeting COSMIN criteria to support its reliability [62].

### AIM-T validity

The structural validity of the AIM-T composed of three subscales, core, lower limb and upper limb, is supported by our results. Indeed, mobility activities may involve the central, lower and

upper body in isolation or in combination, which may result in subdomains of mobility [61,77,78]. Subdomains have also been identified within existing unidimensional measures or item banks, reflecting either mobility versus self-care activities within the BI, the motor FIM or the AM-PAC, or lower versus upper limb activities with the PROMIS-PF, SMFA and motor FIM [79–85]. This study confirms that the AIM-T subscale scores provide more accurate information than the total score, thus supporting their use where feasible.

The AIM-T reflected the expected differences in independence in activities among patients after injury, thereby supporting the AIM-T known-group validity. Specifically, the clear difference between patients with upper versus lower limb injuries is consistent with differences documented by other measures, and further confirms the relevance of the AIM-T subscales [68,86]. The acuteness of injury was discriminated significantly by the AIM-T scores, which is coherent with differences identified by other measures as well as with the expected recovery of independence with time [87–89]. Additionally, the moderate observed difference (d = 0.5) might be related to the heterogeneity of patients as well to some patients having received continued reconstructive care, potentially prolonging their acute phase [90]. The ability of the AIM-T to discriminate acuteness longitudinally needs to be further investigated through responsiveness studies.

The strong correlation with the BI supports the concurrent validity of the AIM-T, and both measures require a similar administration time. The AIM-T's added value is that its activities can be performed similarly across patients, while being considered culturally appropriate to observe. Indeed, while most patients preferred to self-report the BI self-care activities, only few patients refused to perform any of the AIM-T activities, the reasons for which were pain or fear. Moreover, self-care activities may be performed differently across cultures and genders, leading to interpretation difficulties [34,36,91–93]. The AIM-T only includes mobility activities, having a more standardized performance. Additionally, the AIM-T instructions have been complemented with drawings to ease the understanding by patients and limit language barriers. Thus, by including only mobility activities that are culturally appropriate and standardized, comparability of results across settings should be facilitated. However, this may need further investigation since cross-cultural validity of the AIM-T was not assessed in the present study.

## AIM-T reliability

The reliability of the AIM-T for patients after injury was supported for both the total scale and the subscales, through the testing of different measurement properties. This study shows that the items comprised within each AIM-T subscale consistently measure a defined construct, namely the independence in core, lower and upper limb activities [73]. The high values observed for the upper and lower limb subscales ($\alpha > 0.9$) are consistent with other measures assessing similar constructs [87,89,94–96]. The good to excellent inter-rater reliability reaches the COSMIN criteria (ICC>0.70). This is comparable to that of other validated measurements, even though information in similar populations is lacking [79,97,98]. The AIM-T scores from second raters were systematically lower than those of first raters. This might suggest that patients have been too tired to perform the activities similarly or have experienced increased pain during the second assessments. The short time interval was chosen to ensure stability in health status of the patients, as done by others [98]. Moreover, the systematic difference between first and second raters is below the SEM, and therefore falls within the expected true scores.

## Study strengths and limitations

This study has several strengths. Using the COSMIN framework ensured a rigorous methodology that enhances the use of the findings. Testing validity and reliability of a new measure is

part of its development process to ensure its adequacy [73]. To mitigate the risk for confirmation bias, data was collected by trained raters not involved in the study design or data analysis. An additional strength is the inclusion of a range of raters and a heterogenous patient population, including children. Children are often excluded in trauma research despite bearing an important burden of injuries in humanitarian settings [99]. Also, the inclusion of pain assessment during activity and not just at rest allowed reflection of the correlation between pain and activity independence. This is in line with recommendations for assessing pain intensity in acute pain management, such as after surgery or trauma [100–102]. Testing the AIM-T against a widely used and validated measure such as the BI adds to current literature as it has been used in similar settings [48–52,103]. Besides, the BI was chosen as comparator measure due to its brevity, simplicity, and affordability. Lastly, the Cohen's d was used to assess the magnitude of the difference in AIM-T scores between patient subgroups, while others have used the SEM or not specified the magnitude [31,68,87]. Though COSMIN recommends to pre-specify and assess magnitude of differences, there is no guidance on assessment methods, complicating the comparability of results between studies. The use of the effect size allows to assess the magnitude of the differences between groups, as intended in construct validity testing [104].

This study has limitations, some connected to the challenges of conducting research in humanitarian settings. One limitation included the use of four different pairs of raters for assessing inter-rater reliability. This was done to reduce the burden of data collection and to ensure diversity of settings. To mitigate variability between pairs, all raters were trained by the first author using the same training package [97]. Another possible limitation was giving each participant the option to choose one out of two different pain scales (i.e., VAS and FPS-R), based on acceptability of the scales by the different providers [58,105]. There are contradictory findings regarding the strength of the correlation between the pain scales, hindering their interchangeability [57,58,106]. However, since all four correlation coefficients with the AIM-T are coherent, we do not consider this to have a major impact on our findings. Lastly, regarding structural validity, the three-factor model showed a limited goodness of fit, reaching only one fit criterion out of three. However, similar values have been considered acceptable by others, and only one fit criterion is required by COSMIN framework to support structural validity [62,107]. Furthermore, the measure was only tested in MSF-run or -supported health facilities, which could limit external validity. However, this is partly mitigated by the inclusion of four different humanitarian settings, located in different geographical regions.

## Clinical implications and future research

Overall, more high-quality rehabilitation research has been called for, as well as strengthened health information systems, using data on functioning and rehabilitation, including in humanitarian settings [108]. The primary objective when developing the AIM-T was to provide a measure which could be routinely used by any healthcare professionals, quantifying the burden of injury beyond mortality and morbidity, while also being useful to guide clinical practice. Though it has been developed, used and tested within MSF projects, the AIM-T is intended to be useful in any health facilities providing trauma care, and especially where resources are limited. However, health authorities' awareness on the importance of collecting such data within trauma care is required, consistent with the more global call for stronger health information systems in LMICs [19,109,110]. Additionally, to ensure quality of data, a training package has been compiled for healthcare professionals, enhancing the AIM-T reliability and implementation. The package includes a guideline with a detailed description and illustration of each AIM-T activity, as well as a poster displaying the 12 activities and a flowchart explaining the scoring system (S4 Fig). An open access e-learning module in English will

soon be available to enhance access to the training materials, while French and Arabic translations are planned in the near future. Having the SEM and SDC values of AIM-T informs healthcare professionals about its interpretability, but needs to be complemented with other reference values, such as its responsiveness, including the minimum important change. The use of the three subscales is recommended to better capture the independence and multidimensionality among patients after injury. To report the burden of injury, having only one total score to reflect independence in activities is favored, complying with operational needs and constraints. The unidimensionality of the AIM-T total scale was however not supported by our data and should be further investigated in a larger sample, comparing model fit with different profiles of patients. Additionally, we plan to conduct a longitudinal study using the AIM-T as one of the measures to document recovery of functioning after injury in different humanitarian settings and to identify its association with early rehabilitation. Further, describing the experience of healthcare professionals and patients with the use of the AIM-T to encourage early rehabilitation in such settings would be valuable.

## Conclusion

This study supports the validity and reliability of the AIM-T as a measure of independence in activities in patients after injury in humanitarian settings, complementing previous findings on its validity. Thus, the AIM-T could be a valuable measure in trauma care to assess outcomes after injury in humanitarian settings, fostering the reporting of functioning and the monitoring of quality of trauma care in such settings.

## Supporting information

**S1 Fig. Activity Independence Measure–Trauma (AIM-T), the measure.**
(PDF)

**S2 Fig. The three-factor model of the AIM-T with factor loadings.**
(PDF)

**S3 Fig. Bland and Altman plots for the AIM-T subscales.**
(PDF)

**S4 Fig. Activity Independence Measure–Trauma (AIM-T), poster and scoring system flowchart.**
(PDF)

## Acknowledgments

We would like to thank the local teams in each study centre, as well as the coordination and headquarters teams at Médecins Sans Frontières and Humanity & Inclusion. Despite working in challenging environments, their commitment and support made this study happen. More particularly, we would like to thank the eight raters as well as Geraldine Duc, Vincent Lambert, Khaled Ahmedana, Mohammed Abed, Zoe Clift and Jacinth Yan for their valuable contributions to this research project. We would also like to thank our study participants who took their time to participate on the data collection. This study has been funded by Elrha's Research for Health in Humanitarian Crises (R2HC) Programme, which aims to improve health outcomes by strengthening the evidence base for public health interventions in humanitarian crises. R2HC is funded by the UK Foreign, Commonwealth and Development Office (FCDO), Wellcome, and the UK National Institute for Health Research (NIHR).

The following co-authors are part of the AIM-T study group: Gedeon Fabrice Touye, Eric Ndiramiye, Iza Ciglenecki, Annick Antierens, and Arielle Calmejane.

## Author Contributions

**Conceptualization:** Bérangère Gohy, Christina H. Opava, Johan von Schreeb, Rafael Van den Bergh, Aude Brus, Julie Van Hulse, Eric Weerts, Nina Brodin.

**Data curation:** Bérangère Gohy.

**Formal analysis:** Bérangère Gohy, Nina Brodin.

**Funding acquisition:** Bérangère Gohy, Christina H. Opava, Johan von Schreeb, Rafael Van den Bergh, Aude Brus, Julie Van Hulse, Eric Weerts, Nina Brodin.

**Investigation:** Bérangère Gohy, Nicole Fouda Mbarga, Delphine Rougeon, Evelyne Côté Grenier, Lívia Gaspar Fernandes, Eric Weerts.

**Methodology:** Bérangère Gohy, Christina H. Opava, Johan von Schreeb, Rafael Van den Bergh, Aude Brus, Julie Van Hulse, Eric Weerts, Nina Brodin.

**Project administration:** Bérangère Gohy, Rafael Van den Bergh, Jean Patrick Ouamba, Jean-Marie Mafuko, Irene Mulombwe Musambi, Evelyne Côté Grenier, Julie Van Hulse, Eric Weerts, Nina Brodin.

**Supervision:** Bérangère Gohy, Rafael Van den Bergh, Jean Patrick Ouamba, Jean-Marie Mafuko, Irene Mulombwe Musambi, Evelyne Côté Grenier, Julie Van Hulse, Eric Weerts, Nina Brodin.

**Validation:** Bérangère Gohy, Christina H. Opava, Johan von Schreeb, Rafael Van den Bergh, Aude Brus, Julie Van Hulse, Eric Weerts, Nina Brodin.

**Writing – original draft:** Bérangère Gohy, Christina H. Opava, Johan von Schreeb, Rafael Van den Bergh, Nina Brodin.

**Writing – review & editing:** Bérangère Gohy, Christina H. Opava, Johan von Schreeb, Rafael Van den Bergh, Aude Brus, Nicole Fouda Mbarga, Jean Patrick Ouamba, Jean-Marie Mafuko, Irene Mulombwe Musambi, Delphine Rougeon, Evelyne Côté Grenier, Lívia Gaspar Fernandes, Julie Van Hulse, Eric Weerts, Nina Brodin.

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
