## [Decision Letter · Decision Letter 0]

19 Jun 2023

PGPH-D-23-00182

Assessing independence in mobility activities in trauma care: validity and reliability of the Activity Independence Measure-Trauma (AIM-T) in humanitarian settings

Dear Dr. Gohy,

Thank you for submitting your manuscript to PLOS Global Public Health. After careful consideration, we feel that it has merit but does not fully meet PLOS Global Public Health’s publication criteria as it currently stands. Therefore, we invite you to submit a revised version of the manuscript that addresses the points raised during the review process.

We look forward to receiving your revised manuscript.

Kind regards,

Vinay Nair Kampalath, MD, DTMH

Guest Editor

Journal Requirements:

1. We noticed you have some minor occurrence of overlapping text with the following previous publication(s), which needs to be addressed:

- https://doi.org/10.1371/journal.pgph.0001334

In your revision ensure you cite all your sources (including your own works), and quote or rephrase any duplicated text outside the methods section. Further consideration is dependent on these concerns being addressed."

Additional Editor Comments (if provided):

Please do address the comments from the two reviewers who recommend minor revisions, particularly in regards to the discussion.

Reviewers' comments:

Reviewer's Responses to Questions

**Comments to the Author**

1. Does this manuscript meet PLOS Global Public Health’s publication criteria? Is the manuscript technically sound, and do the data support the conclusions? The manuscript must describe methodologically and ethically rigorous research with conclusions that are appropriately drawn based on the data presented.

Reviewer #1: Yes

Reviewer #2: Yes

2. Has the statistical analysis been performed appropriately and rigorously?

Reviewer #1: Yes

Reviewer #2: Yes

3. Have the authors made all data underlying the findings in their manuscript fully available (please refer to the Data Availability Statement at the start of the manuscript PDF file)?

Reviewer #1: No

Reviewer #2: Yes

4. Is the manuscript presented in an intelligible fashion and written in standard English?

Reviewer #1: Yes

Reviewer #2: Yes

5. Review Comments to the Author

Reviewer #1: Summary

The authors of this manuscript conduct a validity and reliability assessment of a score to assess functional outcomes after injury for use in humanitarian settings. In doing so, they address a significant gap in humanitarian care. A recent systematic review of civilian conflict casualties found that only approximately 1% of reports presented any data on functional outcomes other than mortality. Therefore, their manuscript represents a significant contribution. Their methods are robust, quality of writing clear, and discussion pertinent to broader dialogues. This report adds value to academic dialogue around rehabilitation and quality of life after trauma in low-resource/humanitarian settings and merits publication in PLOS Global Public Health with minor revisions though some revisions are recommended prior to meeting publication standards.

Recommendations/Comments

- Introduction: clearly written, appropriately contextualizes the gap addressed by the authors manuscript (e.g. importance of a functional disability metric appropriate to humanitarian settings). They do an excellent job of highlighting the limitations of existing scores (e.g. cannot be implemented in settings of limited literacy, overly lengthy, culturally inappropriate.) The only change I would recommend is that, as written, it is slightly unclear whether this is the first time the AIM-T is being presented as a score, or whether the AIM-T has previously been described and the authors here are conducting a validity assessment. Would recommend they provide clarification.

- General: minor proofreading is required throughout the manuscript (e.g. “…centres supported or ran by…” should read “centres supported or run by,” page 6)

- Methods: well organized and well presented. Although this is minor, the authors select a very odd and obscure reference to discuss their use of the SATS as a triage score. They should (a) cite this the first time it is used (currently the reference is appended to the second sentence) and (b) select a more relevant reference regarding the SATS. Only other additional comment is that the methods section seems unnecessarily long. The authors could consider making this more concise, particularly the section describing the various measurements scores and procedures which contain excessive detail that makes the manuscript unwieldy.

- Results: age of raters presented in the demographics section does not seem relevant

- Discussion: Overall this is the section that requires the most reconsideration. On page 20, recommend the authors reword this paragraph. As written, it is unclear what they propose the AIM-T to offer over the BI after they raise the point that added value is equivocal. This sentence is confusing and unclear as written: “Thus, by including only culturally acceptable activities, the AIM-T aims to be composed of activities performed similarly across settings, thereby promoting early mobilization while also facilitating comparability of results across settings..” Recommend the authors restructure the final paragraphs of the section. It is unconventional to have a section “strengths and limitations” at the end of the discussion as opposed to just limitations. The discussion would be more compelling and coherent if contexualization of its strengths were integrated with the discussion of its validity/reliability, and just a short paragraph on limitations prior to conclusion were presented in the standard fashion. The authors should also restructure the discussion section such that “implications and future research” is presented prior to the limitations paragraph.

- Conclusion: should contain a sentence clearly stating what the AIM-T adds to prior research

Reviewer #2: I would like to express my gratitude to the editor for giving me the opportunity to review this manuscript, which I enjoyed reading.

The authors present a study as part of a larger project of considerable importance, focusing on a highly relevant topic and research gap. The study design and analysis are robust and demonstrate the authors' expertise in the field. The authors should be commended for conducting such a study in multiple challenging settings. The inclusion of children and thus the perspective to apply the AIM-T across ages is a particular strength in the design. The AIM-T score seems simple and comprehensive at the same time and it is nicely illustrated in Fig 4. Furthermore, the diversity of the authors is worth highlighting.

There are some points in the discussion that should be strengthened, and I have also noted minor comments throughout the manuscript

- Page 4: the link between ‘burden of injury’ and ‘people living with disability’ is not clear. Both groups are not necessarily the same, especially if the authors write ‘persons living with disability’ rather than something like ‘persons with trauma-caused disability’. A sentence explaining the link between injury and resulting impairment or certain forms of disability would help.

- Page 5: lack of language validation of outcome measures is another barrier in addition to lack of cultural validation

- Page 5: “for clinicians to mobilize patients earlier” – it may help to define which professional groups are supposed to apply the measure

- “It was purposively developed as an observed measure to encourage patients to move early on in their care process and for clinicians to mobilize patients earlier, both of which are known challenges in humanitarian settings” – this sentence seems a bit out of place; outcome measurement development and changing patient/clinician behaviour around early mobilization are two different concepts; outcomes measures should be developed to measure outcomes, see also comment below re discussion;

- Page 7: participate in the study (not ‘to’)

- Page 9, procedure: In which language(s) was the training conducted? Did the trainees have sufficient command of these, and were these the languages in which they would then explain the AIM-T to participants/patients? How did the authors ensure that the information was not lost in translation?

- Page 13: please add some detail on most common injuries.

I would like to see some detail in the discussion on

- Language and (health) literacy issues: the authors do not mention anywhere in the manuscript

o If they faced language barriers in the use of the BI and which language versions were used, respectively;

o If they faced language barriers during trainings and application of the AIM-T and how to overcome these, see comment re page 9

o in Clinical implications: Training package / e-learning module sound very valuable, are these delivered in English, French, Arabic, local languages? Will they be available outside the INGO world? Please add

- It seems as if AIM-T was developed for settings where INGOs lead or support projects. A measure like the AIM-T is crucial for any fragile settings and it should not be applied only in the presence of INGOs. The question is how it can be of help for local structures in fragile systems and how it can be transferred to local knowledge to enhance standards and practice around outcome measurement. The discussion/ limitations sections should include some reflections on this.

- Clinical implications and future research :

o “Although feedback from the field teams is supportive, it is still to be studied whether the AIM-T actually promotes early mobilization at patient and clinician level.” – Although early mobilization should indeed be promoted, I am not convinced the development of a measure will or should have that purpose; A measure designed for humanitarian settings should allow outcomes to be validly and routinely measured - a well known barrier- to determine the impact of early mobilisation, prevention of permanent impairment etc

o the overall research project includes various steps in validation and use of the AIM-T, which is excellent, and the authors refer to a subsequently planned study. They are hereby addressing a hugely underresearched subject. However, I would also like to see some reflection on future research needs in the sector and in humanitarian settings, such as studies on overall rehabilitation impact measurement, rehabilitation outcome measurement development / cultural + language validation, and more.

- References: 38 incomplete

- S4_Fig: Is there a reason why the activity to get up from the floor into standing and back is not part of the AIM-T? In many humanitarian contexts, this is a typical daily function.

6. PLOS authors have the option to publish the peer review history of their article (what does this mean?). If published, this will include your full peer review and any attached files.

**Do you want your identity to be public for this peer review?** For information about this choice, including consent withdrawal, please see our Privacy Policy.

Reviewer #1: No

Reviewer #2: No

---

## [Decision Letter · Decision Letter 1]

18 Aug 2023

Assessing independence in mobility activities in trauma care: validity and reliability of the Activity Independence Measure-Trauma (AIM-T) in humanitarian settings

PGPH-D-23-00182R1

Dear Mrs Gohy,

We are pleased to inform you that your manuscript 'Assessing independence in mobility activities in trauma care: validity and reliability of the Activity Independence Measure-Trauma (AIM-T) in humanitarian settings' has been provisionally accepted for publication in PLOS Global Public Health.

Best regards,

Vinay Nair Kampalath, MD, DTMH

Guest Editor

Thank you for this important contribution, and for addressing the comments from the two reviewers. There are additional minor suggestions/comments from Reviewer 2 on Revision 1.

Reviewer Comments (if any, and for reference):

Reviewer's Responses to Questions

**Comments to the Author**

1. If the authors have adequately addressed your comments raised in a previous round of review and you feel that this manuscript is now acceptable for publication, you may indicate that here to bypass the “Comments to the Author” section, enter your conflict of interest statement in the “Confidential to Editor” section, and submit your "Accept" recommendation.

Reviewer #1: All comments have been addressed

Reviewer #2: (No Response)

2. Does this manuscript meet PLOS Global Public Health’s publication criteria? Is the manuscript technically sound, and do the data support the conclusions? The manuscript must describe methodologically and ethically rigorous research with conclusions that are appropriately drawn based on the data presented.

Reviewer #1: Yes

Reviewer #2: Yes

3. Has the statistical analysis been performed appropriately and rigorously?

Reviewer #1: Yes

Reviewer #2: Yes

4. Have the authors made all data underlying the findings in their manuscript fully available (please refer to the Data Availability Statement at the start of the manuscript PDF file)?

Reviewer #1: No

Reviewer #2: Yes

5. Is the manuscript presented in an intelligible fashion and written in standard English?

Reviewer #1: Yes

Reviewer #2: Yes

6. Review Comments to the Author

Reviewer #1: All reviewer comments have been adequately addressed. The authors' contributions to evaluating functional outcomes after injury in humanitarian settings are appreciated.

Reviewer #2: Thank you for revising the manuscript and considering my suggestions and comments.

Most comments have been clearly addressed. A few comments require more detail and a few sentences and expressions may need corrections.

• Overall, I found it difficult to follow the point-by-point response and the respective place in the manuscript as the authors provide no line or page reference. The track changes format is not very suitable for reading flow and doesn’t show which reviewer's comment is addressed at which point in the text and what other changes were made independently from reviewer comments. It would be important if the authors could signpost reviewers to the respective lines (in different colour, but not in track changes) in the manuscript.

• I suggest to reformulate the last sentence of the first paragraph in the Introduction, page 5 (track changes) into: “As early mobilisation in particular is a common challenge in humanitarian settings, the use of outcome measures for daily functioning may encourage increased implementation and promotion of these activities as part of routine care.”

• change to “any healthcare professional” or “all healthcare professionals” in the text

• Idem for “any health facility” or “all health facilities”

• I don’t see a specific reference to local stakeholder involvement which would be relevant to add.

• The future research section has been revised with an additional sentence on research/ strengthened HIS needs. However, I would appreciate some discussion addressing my comment, specifically with regards to “reflection on future research needs in the sector and in humanitarian settings, such as studies on overall rehabilitation impact measurement, rehabilitation outcome measurement development / cultural + language validation, and more.”

• Page 23, bottom:

• Please add a phrase and reference regarding the introduction of functioning as a third health indicator;

• ‘guide clinical practice’ was not the primary objective, better to be placed in an extra sentence;

• I suggest to reformulate, e.g. into: “---which could be routinely used by any healthcare professional, thus consolidating the use of "functioning" as a health indicator beyond mortality and morbidity, as discussed and proposed in various publications (REF). Furthermore, the AIM-T may also be useful to enhance clinical practice.”

• Typo p 27: change to “Acknowledgements”

7. PLOS authors have the option to publish the peer review history of their article (what does this mean?). If published, this will include your full peer review and any attached files.

**Do you want your identity to be public for this peer review?** For information about this choice, including consent withdrawal, please see our Privacy Policy.

Reviewer #1: No

Reviewer #2: No
